# ERK1/2 Inhibition via the Oral Administration of Tizaterkib Alleviates Noise-Induced Hearing Loss While Tempering down the Immune Response

**DOI:** 10.3390/ijms25126305

**Published:** 2024-06-07

**Authors:** Richard D. Lutze, Matthew A. Ingersoll, Alena Thotam, Anjali Joseph, Joshua Fernandes, Tal Teitz

**Affiliations:** 1Department of Pharmacology and Neuroscience, School of Medicine, Creighton University, Omaha, NE 68178, USA; richardlutze@creighton.edu (R.D.L.); matthew.ingersoll@nih.gov (M.A.I.); alenathotam6@gmail.com (A.T.); anjalijoseph@creighton.edu (A.J.); jfernandes@kumc.edu (J.F.); 2The Scintillon Research Institute, San Diego, CA 92121, USA

**Keywords:** ERK1/2, hearing protection, immune response, repurposing drugs, oral delivery, noise-induced hearing loss

## Abstract

Noise-induced hearing loss (NIHL) is a major cause of hearing impairment and is linked to dementia and mental health conditions, yet no FDA-approved drugs exist to prevent it. Downregulating the mitogen-activated protein kinase (MAPK) cellular pathway has emerged as a promising approach to attenuate NIHL, but the molecular targets and the mechanism of protection are not fully understood. Here, we tested specifically the role of the kinases ERK1/2 in noise otoprotection using a newly developed, highly specific ERK1/2 inhibitor, tizaterkib, in preclinical animal models. Tizaterkib is currently being tested in phase 1 clinical trials for cancer treatment and has high oral bioavailability and low predicted systemic toxicity in mice and humans. In this study, we performed dose–response measurements of tizaterkib’s efficacy against permanent NIHL in adult FVB/NJ mice, and its minimum effective dose (0.5 mg/kg/bw), therapeutic index (>50), and window of opportunity (<48 h) were determined. The drug, administered orally twice daily for 3 days, 24 h after 2 h of 100 dB or 106 dB SPL noise exposure, at a dose equivalent to what is prescribed currently for humans in clinical trials, conferred an average protection of 20–25 dB SPL in both female and male mice. The drug shielded mice from the noise-induced synaptic damage which occurs following loud noise exposure. Equally interesting, tizaterkib was shown to decrease the number of CD45- and CD68-positive immune cells in the mouse cochlea following noise exposure. This study suggests that repurposing tizaterkib and the ERK1/2 kinases’ inhibition could be a promising strategy for the treatment of NIHL.

## 1. Introduction

Hearing loss afflicts more than 10% of the world population, with noise-induced hearing loss (NIHL) as one of its main causes [1,2,3,4]. NIHL has increased in recent years due to the use of personal headphones, noisy city environments, and noise exposure in military combat [5,6,7,8,9]. Recent studies show that NIHL is linked to dementia and mental health conditions such as depression [10,11]. Hearing loss occurs with intense noise exposure due to synaptopathy, the mechanical damage to stereocilia and cellular mechanisms that leads to cell death and dysfunction [12,13,14,15]. Inhibiting the pathways that induce cochlear stress and dysfunction is a promising approach to limit the amount of hearing loss that occurs with damaging noise exposure [13,16]. There are currently no Food and Drug Administration (FDA)-approved drugs for the treatment of NIHL; therefore, there is a clinical need to develop compounds that can protect individuals from this very common disorder [4,17,18].

The mitogen-activated protein kinase (MAPK) pathway is involved in a multitude of cellular processes and is a common target for many diseases [19]. The MAPK pathway is commonly deregulated in various types of cancer but it has been implicated in hearing loss over the last 15–20 years [4,20,21,22,23]. This cellular pathway consists of a phosphorylation cascade of different proteins that are activated when phosphorylated. ERK1/2 is the main kinase in this cascade, directly downstream of the kinases RAF and MEK, and activates downstream pathways and transcription factors [24]. Activation of the MAPK pathway has traditionally been associated with cell proliferation and survival but this is dependent on the cell type and stimulus [19,24]. Cells in the inner ear are post-mitotic and not actively proliferating, and several studies have demonstrated that the activation of this pathway in these types of tissues can cause cellular stress, dysfunction, and ultimately, cell death [4,21,25,26,27,28]. Due to these findings in post-mitotic cochlear cells and other tissues, targeting ERK1/2 is a promising approach to mitigating NIHL. 

Tizaterkib (formerly known as AZD-0364) is a newly developed, orally bioavailable, highly specific ERK1/2 inhibitor that is currently in phase 1 clinical trials for the treatment of advanced solid tumors and hematological malignancies [29,30]. This compound was specifically synthesized to have a high specificity to ERK1/2 and to limit any off-target binding. Additionally, the drug was shown to have 30% oral bioavailability in rats and 100% in dogs [30]. It was demonstrated to have a low IC_50_ in vitro of 6 nM and a high affinity for the ERK1/2 kinases [30]. Furthermore, our laboratory has shown that tizaterkib protects cochlear explants from cisplatin-induced outer hair cell (OHC) death with an IC_50_ of 5 nM, close to the IC_50_ value of 6 nM measured for the inhibition of ERK1/2 kinase activity in cell lines [4,30]. Due to the high specificity of the drug for inhibiting ERK1/2 and its low predicted doses that will limit off-target effects, tizaterkib is a very useful tool to study ERK1/2’s role in NIHL and is promising for repurposing as an otoprotective drug. Targeting ERK directly in the MAPK pathway may be more efficient for protection from hearing loss because ERK modulates different downstream pathways that have been implicated in mechanisms that lead to NIHL, such as cellular death and inflammation [24,28,31,32]. Additionally, our laboratory demonstrated that dabrafenib, a BRAF kinase inhibitor, protects mice from NIHL [4]. The use of an ERK inhibitor would confirm that the inhibition of the canonical MAPK pathway, not just the upstream molecular target BRAF, mitigates NIHL. 

One cellular mechanism that is associated with NIHL is the induction of the inflammatory response, which produces an immune response in the cochlea [13,33,34,35,36,37]. Several studies have shown that the inhibition of cytokines and chemokines protects mice from NIHL and other types of hearing loss [38,39,40,41,42]. Additionally, dexamethasone, a corticosteroid which acts as an anti-inflammatory agent, has been shown to be protective against many types of hearing loss, including NIHL [43,44]. Furthermore, infiltrating immune cells from the peripheral blood have been implicated in the damage to cochlear cells as a result of a bystander effect in which immune cells release cytokines and chemokines, which exacerbate that inflammatory response and can cause cellular dysfunction and eventually cell death [34,37,45,46]. This suggests that the inflammatory response, which leads to an overactive immune response in the cochlea, is partially responsible for the damage that occurs after noise insults [47]. The MAPK pathway has been shown to regulate the inflammatory and immune response and one of the possible ways that MAPK inhibition protects from NIHL is through modulating this critical cellular response following noise [25,48,49,50,51,52]. 

In this study, we wanted to elucidate whether an ERK1/2 inhibitor protects mice from permanent hearing loss, which has not been demonstrated in vivo. Additionally, we wanted to demonstrate that tizaterkib can safely be administered, which would be useful for not just hearing protection, but for many other neuroinflammatory diseases in which ERK1/2 dysregulation/activation occurs as well. In this study, we perform a dose–response measurement of tizaterkib in an animal model to determine the minimum effective oral dose that protects from NIHL. We examine whether ERK1/2 inhibition mitigates the noise-induced synaptopathy which commonly occurs with damaging noise exposures, and we demonstrate that tizaterkib protects from NIHL primarily through the MAPK pathway by employing the KSR1 KO genetic mouse model [53,54,55]. We also show that tizaterkib mitigates NIHL not only after 2 hours of noise exposure to 100 dB SPL, but also at a higher noise exposure level of 106 dB SPL for 2 h, which widens the therapeutic potential of the drug. Furthermore, we carry out different schedules of administration of the drug to determine the optimal treatment regimen to protect from NIHL through MAPK inhibition. Finally, we test whether ERK1/2 inhibition alters the immune response following noise exposure to elucidate one of tizaterkib’s mechanisms of protection from NIHL. 

## 2. Results 

### 2.1. Tizaterkib Protects from Noise-Induced Hearing Loss When Treatment Begins 45 min before Noise Exposure

We first determined whether tizaterkib (chemical structure in Figure 1A) protects from NIHL when mice were pretreated with the drug. Briefly, mice were orally treated with 25 mg/kg/bw tizaterkib, a dose previously published as having no side effects for mice [29,30], 45 min before noise exposure. Mice were exposed to 100 dB SPL noise for 2 h at the 8–16 kHz octave band. This noise exposure induces permanent threshold shifts in FVB/NJ mice [4,17]. Mice were then orally treated again in the evening and treatment proceeded for another 2 days, for a total of 3 days of treatment, twice a day (Figure 1B). Mice treated with tizaterkib had significantly lower Auditory Brainstem Response (ABR) threshold shifts at the 8, 16, and 32 kHz frequencies compared to the noise-alone cohort. Tizaterkib-treated mice had an average reduction in their ABR threshold shifts of 27 dB at 8 kHz, 16 dB at 16 kHz, and 15 dB at 32 kHz compared to the noise-alone mice (Figure 1C,D). Tizaterkib-alone-treated mice had no hearing loss, which indicates the drug causes no ototoxicity when administered by itself. In addition, the mice treated with 25 mg/kg/bw twice a day for three days did not display any signs of general toxicity, as determined by weight measurements compared to carrier-alone-treated mice (Figure 1E). 

### 2.2. Tizaterkib Administration Protects from NIHL When Treatment Starts 24 h after Noise Exposure

Noise exposure cannot always be predicted; therefore, we determined whether protection still occurs when the treatment begins after noise exposure. Figure 2A shows the treatment and noise exposure protocol in which the tizaterkib treatment began 24 h after noise exposure and mice were orally treated twice a day for three total days. A post-noise exposure treatment of 25 mg/kg tizaterkib significantly lowered the ABR threshold shifts compared to noise-alone-treated mice. An average threshold shift reduction of 20 dB at 8 kHz, 25 dB at 16 kHz, and 10 dB at 32 kHz was observed compared to the noise-alone cohort (Figure 2B). Furthermore, 5 mg/kg tizaterkib conferred the same amount of protection as 25 mg/kg (Figure 2B). A lower dose of 0.5 mg/kg was then tested, which also significantly lowered the ABR threshold shifts at the 8 and 16 kHz frequencies. An average threshold shift reduction of 22 dB at 8 kHz and 23 dB at 16 kHz was observed compared to noise-alone-treated mice (Figure 2C). A dose of 0.1 mg/kg tizaterkib was then tested to determine the minimum effective dose and 0.1 mg/kg did not offer significant protection, which indicates that 0.5 mg/kg was close to the minimum effective dose needed to protect from NIHL (Figure 2D). Figure 2E shows a dose–response curve of the protection from NIHL at the 16 kHz frequency, with 25, 5, and 0.5 mg/kg all offering similar levels of protection. Males and females were analyzed separately to test whether there were any sex differences, and tizaterkib equally protects both sexes from NIHL (Figure 2F) (Figure 2G and Appendix A). 

### 2.3. Tizaterkib Protects from Noise-Induced Cochlear Synaptopathy 

Following the post-treatment hearing tests, mouse cochleae were collected and stained with Ctbp2 and myosin VI to determine whether ERK1/2 inhibition protects from cochlear synaptopathy, which occurs following noise exposure [4,17,56]. In this mouse model of permanent hearing loss utilizing FVB/NJ mice, no outer hair cell loss occurs; therefore, we measured synaptopathy, which contributes to the hearing loss phenotype in these mice [4,17,57]. Representative images of cochlear samples stained with Ctbp2 and myosin VI are shown in Figure 3A,C, which represent the 8 and 16 kHz regions, respectively. The number of ctbp2 puncta per inner hair cell (IHC) were quantified for the 8 and 16 kHz regions (Figure 3B,D). The average number of Ctbp2 puncta per IHC in the 8 and 16 kHz regions in non-noise-exposed mice was 12.3 and 14.2, respectively. The noise-alone cohort had an average of 9.1 and 9.7 ctpb2 puncta per IHC in the 8 and 16 kHz regions, respectively. The 0.5 mg/kg tizaterkib-treated mice had significantly more ctbp2 puncta compared to noise-alone mice, with 11.9 at 8 kHz and 11.5 at 16 kHz (Figure 3B,D). Additionally, tizaterkib-treated mice had a significantly higher ABR wave 1 amplitude at 16 kHz following noise exposure compared to noise-alone mice at 90 dB SPL (Figure 3E).

### 2.4. Tizaterkib Protects Mice from NIHL When Exposed to a Higher Noise Exposure Intensity of 106 dB SPL 

In this experiment, the same drug treatment protocol as before was followed, except mice were exposed to 106 dB SPL noise instead of 100 dB SPL (Figure 4A). The noise-alone group had, on average, 10 dB higher ABR threshold shifts compared to mice exposed to 100 dB (Figure 2). Mice orally treated with 0.5 mg/kg tizaterkib had significantly lower ABR threshold shifts in the 8 and 16 kHz regions (Figure 4B,C). Tizaterkib-treated mice had lower average threshold shifts of 15 dB at 8 kHz and 22 dB at 16 kHz compared to noise-alone mice. Additionally, tizaterkib-treated mice, following noise exposure, have significantly lower DPOAE threshold shifts at 16 kHz, with an average reduction of 14 dB SPL compared to noise-alone mice (Figure 4D). 

### 2.5. Tizaterkib Treatment in KSR1 KO Mice Does Not Confer Any Extra Protection against NIHL

Tizaterkib was expected to protect mice from NIHL through ERK inhibition, and this was tested by employing KSR1 genetic germline knockout (KO) and wild-type (WT) mouse littermates [55]. KSR1 is a scaffolding protein in the MAPK pathway and eliminating the KSR1 protein significantly lowers MAPK activation and activity (Figure 5A) [53,54,55]. KSR1 WT and KO littermate mice, C57BL/6 strain, were exposed to 100 dB SPL noise, in the 8–16 kHz octave band, for 2 h and then treated via oral gavage with tizaterkib or carrier twice a day for three days, starting 24 h post noise exposure (Figure 5B). The tizaterkib treatment in KSR1 WT mice offers significant protection from NIHL at 16 and 32 kHz, with an average reduction in ABR threshold shifts of 18 and 19 dB, respectively (Figure 5C). KO mice alone and KO mice treated with tizaterkib have an almost identical protection from NIHL, with average reductions in their threshold shifts of 27 dB at 16 kHz and 28 dB at 32 kHz. There is no significant difference in threshold shifts between the WT mice treated with tizaterkib and either KO cohort exposed to noise (Figure 5C). 

### 2.6. Three Days of Oral Tizaterkib Treatment Produces Better Protection from NIHL Compared to One and Two Days of Treatment 

Mice treated with 0.5 mg/kg tizaterkib twice a day for three days had significant protection from NIHL when the treatment begins 24 h after noise exposure. We checked whether a delay in the first treatment to 48 h following noise exposure would still offer significant protection. We performed the same noise and tizaterkib treatment protocol, except the first tizaterkib treatment occurred 48 h after noise and not 24 h (Figure 6A). Mice treated with 0.5 mg/kg tizaterkib did not have significantly lower ABR threshold shifts compared to the noise-alone mice, even though they did trend lower (Figure 6B). We then determined whether a single day or 2 days of treatment instead of 3 days, starting 24 h after noise, offers similar protection (Figure 6C). Mice treated with 0.5 mg/kg tizaterkib for one day did have significantly lower threshold shifts at 32 kHz and mice treated with tizaterkib for 2 days had lower threshold shifts at 8 and 16 kHz with slightly better protection than after 1 day of treatment (Figure 6D). Three days of treatment was significantly better than one and two days of treatment. Compared to mice treated for 2 whole days, mice treated for 3 whole days had average lower threshold shifts of 12, 15, and 11 dB in the 8, 16, and 32 kHz regions, respectively (Figure 6D). 

### 2.7. Tizaterkib-Treated Mice Have Significantly Less Immune Cells in Their Cochleae Compared to Noise-Alone-Treated Mice 

Modulating the number of immune cells in the cochlea following noise insult could lead to protection from NIHL, and ERK1/2 has been demonstrated to modulate that immune response [58,59,60]. We exposed the mice to the same noise as before (100 dB for 2 h in the 8–16 kHz octave band) and treated the mice with carrier or 0.5 mg/kg tizaterkib for three days, twice a day. Mice were then sacrificed 1 h after the last drug treatment and stained with anti-CD45 antibody to determine the number of total immune cells in the cochlea following noise insult. Mice exposed to noise and treated with carrier had a significant increase in the number of CD45-positive cells in their cochleae, while the tizaterkib treatment significantly lowered the number of CD45-positive cells compared to noise alone to almost carrier or drug-alone levels (Figure 7A,B). The scala tympani region was also analyzed further because this region had the largest increase in immune cells following noise insults. The number of CD45-positive cells in the walls of the scala tympani was significantly lower in tizaterkib-treated mice exposed to noise compared to noise-alone mice and reached the levels of the carrier or drug-alone mice (Figure 7C,D). There was a 1.8-fold difference in CD45-positive cells between the noise-alone- and the noise + tizaterkib-treated mice (Figure 7B) and an even larger fold-difference of 3.2 in CD45-positive cells in the scala tympani region between the noise- and noise + tizaterkib-treated mice (Figure 7D). Additionally, there was a significant 30% decrease in CD45-positive cells in the stria vascularis of tizaterkib-treated mice following noise exposure compared to the noise-alone cohort (Figure 7E,F). No difference was observed in the organ of the Corti region or the spiral ganglion neurons between all groups. Cochlear cryosections stained with secondary antibody alone (no CD45 primary antibody) did not have positive CD45 staining (Appendix A) and normal immune cell morphology was observed when higher-magnification images were taken of CD45-positive cells (Appendix A). 

Total cochlear protein lysates were prepared from mice sacrificed 6 days following noise exposure to examine the duration for which this difference in immune cell numbers persists. Western blots were run and probed with anti-CD45 antibody (general immune cells marker), anti-CD68 antibody (a macrophage marker), and anti-GAPDH antibody (a loading control). Mice treated with tizaterkib had a statistically significant 61% decrease in the amount of CD45 in their cochleae compared to noise-alone mice (Figure 8A,B). The drug-treated mice also had a statistically significant 64% decrease in their amount of CD68 following noise exposure compared to noise-alone mice (Figure 8A,C). For both CD45 and CD68, mice treated with tizaterkib following noise exposure had the same levels as carrier and drug-alone-treated mice (Figure 8B,C). When mice were sacrificed 8 days after the noise insult instead of 4 or 6 days, and cochlear cryosections were stained with CD45, the number of immune cells following noise exposure returned to non-noise expose levels (Appendix A).

## 3. Discussion

Tizaterkib mitigates NIHL at a low dose of 0.5 mg/kg/bw when oral delivery begins 24 h after noise exposure and mice are treated twice a day for three days. A total of 1 mg/kg/bw is administered per day, which is the mouse equivalent to what humans are currently receiving in phase 1 clinical trials for cancer treatment [61]. Tizaterkib has a wide therapeutic window of at least 50 in mice for the treatment of NIHL—25 mg/kg offers significant protection with no known deleterious side effects and 0.5 mg/kg offers an almost identical protection, which makes the therapeutic window at least 50 in mice. A wide therapeutic window is necessary for any drug to make it to clinical trials [16,62,63,64]. These preliminary data on tizaterkib are promising because they demonstrate that ERK1/2 can be targeted for protection from hearing loss with no known side effects, which has been a legitimate concern with ERK1/2 inhibitors in the past.

Tizaterkib protects from NIHL when the drug is first administered 45 min before noise exposure and treatment continues after noise exposure. The drug offers identical protection when the treatment begins 24 h after noise exposure compared to the treatment beginning 45 min before the insult. This is encouraging because noise exposure cannot always be predicted, so the treatment beginning after noise exposure is more translationally relevant and widens the therapeutic applications of targeting the MAPK pathway. Additionally, 3 days of treatment was shown to offer significantly better protection compared to 1 and 2 days of treatment. These treatment timing experiments demonstrate the optimal times to target the MAPK pathway after noise exposure and show the critical times to intervene with treatment in order to grant protection from NIHL. Translationally, tizaterkib was administered to patients in phase 1 clinical trials for 21 days and we generated a significant protection from hearing loss with only 3 days of treatment at the same daily dose [61]. This makes tizaterkib a promising preclinical compound for the treatment of NIHL. The total amount of drug that is given to mice is less than what is currently being administered to humans, which is very crucial when repurposing drugs for the treatment of noise-induced hearing loss. 

ERK1/2 inhibition not only protects from permanent NIHL, but also protects mice from cochlear synaptopathy, which commonly occurs after damaging noise exposures [4,17,65]. Cochlear synaptopathy is when the synapses between the IHC and auditory nerve are damaged and reduced neural transmission occurs. This can be part of the mechanism that causes hearing loss to occur and synaptopathy is also a risk factor for future age-related hearing loss [66,67,68,69,70]. Tizaterkib-treated mice have more Ctbp2 puncta per IHC compared to noise-alone-treated mice, which demonstrates that less synaptic dysfunction occurs with ERK1/2 inhibition. Additionally, tizaterkib-treated mice have larger ABR wave 1 amplitudes, which is a functional correlate of cochlear synaptopathy [71]. The combination of these two results demonstrates the significant protection from synaptic dysfunction in tizaterkib-treated mice compared to noise-alone mice. Part of the protective effect that occurs due to ERK1/2 inhibition could be through the prevention of synaptic dysfunction [4,17,55]. Tizaterkib-treated mice have less synaptic dysfunction and permanent hearing loss compared to noise-alone mice, so the protection of their synapses could be leading to protection from permanent hearing loss. There is a correlation between the number of ribbon synapses and hearing loss in FVB mice; protection from ribbon synapse loss protects from hearing loss [4,17,57,72]. Additionally, the number of orphan synapses has been shown to be minimal 2 weeks following noise damage, which makes Ctbp2 counts a proper measurement of synaptopathy [57,73,74]. 

Tizaterkib protected mice from NIHL at a low dose of 0.5 mg/kg when mice were exposed to 100 dB SPL for two hours; therefore, we wanted to test a higher noise exposure level to examine whether ERK inhibition also protects mice from higher noise levels. Exposing FVB mice to 100 dB for 2 h induces permanent hearing loss, but we wanted to test the drug’s ability to protect against a more intense noise exposure level [4,17,55]. This study demonstrates that tizaterkib protects mice from NIHL when the animals are exposed to 106 dB SPL for two hours. The level of protection seen following 106 dB SPL noise exposure is 90% of the 100 dB 2 h noise protection level (Figure 2C and Figure 4B). We also show that the DPOAE threshold shifts were lower with the tizaterkib treatment, which suggests ERK inhibition protects OHCs from noise-induced dysfunction. Even though no OHC loss occurs, there is still some OHC dysfunction as demonstrated by the increased DPOAE thresholds in noise-exposed mice. OHCs are one of the main cell types in the inner ear affected by noise exposure so this is important information to demonstrate [13,75,76,77]. Protection from a more intense noise exposure demonstrates the therapeutic advantage of targeting the MAPK pathway for the mitigation of NIHL. Previous studies have shown that some drugs protect from 100 dB noise exposures but not 106 dB, which makes targeting this pathway even more attractive and promising [17]. 

The KSR1 mouse model was utilized to demonstrate that tizaterkib was protecting mice through the inhibition of the MAPK pathway and not through some other nonspecific off-target effects. We have recently shown that KSR1 KO mice have a reduced phosphorylation of ERK1/2 in their cochleae after noise exposure and are partially resistant to noise-induced hearing loss compared to their WT KSR1 littermates [55]. Here, we show that the pharmacological inhibition of the molecular target ERK1/2 achieves similar resistance levels to those of the KSR1 KO mice. If the protective effect of tizaterkib was occurring through an off-target effect, we would expect that KSR1 KO mice treated with tizaterkib would have a difference in their protection from NIHL compared to KSR1 KO mice not treated with the drug. Both KO mouse groups, KO alone and KO plus tizaterkib, had identical protection to one another, and WT KSR1 mice treated with the drug were not significantly different from either KO group. This supports our hypothesis that tizaterkib is protecting mice from NIHL through the inhibition of the MAPK pathway, which was expected due to the drug’s specificity for ERK1/2, but nonetheless needed to be tested [29,30]. In the past, a main concern with MAPK inhibitors was their off-target effects and toxicity profiles [78], but this result demonstrates that tizaterkib is not protecting against hearing loss through off-target effects but directly through the inhibition of ERK1/2 and the MAPK pathway. 

Immune cell infiltration to the cochlea following noise exposure has been implicated as a possible mechanism that leads to hearing loss [36,42,79]. This study demonstrates that ERK1/2 inhibition lowers the number of infiltrating immune cells following noise insults. We used CD45 as a general marker for all immune cells to determine whether ERK inhibition affected the entire immune response following noise and not a specific subset of immune cells. Previous studies have shown that the increase in CD45-positive cells is due to infiltrating immune cells and not the proliferation of resident macrophages [45]. In noise-exposed mice, we observed significant increases in CD45-positive cells in the walls of the scala tympani and the stria vascularis. This is in agreement with previous studies that have shown increases in immune cells in the same regions of the cochlea as we observed [45,79,80,81]. Limiting the number of immune cells could protect cochlear cells from the secondary damage that can occur as a result of these infiltrating immune cells [37,42]. 

In our study, we first chose to look at the number of CD45-positive cells in the cochlea 4 days after noise exposure, because previous reports show that the peak in immune infiltrates occurs at this time point [37]. We saw a significant difference between the noise-alone mice and the mice treated with tizaterkib. Additionally, we still observed more CD45 protein in noise-exposed cochlea 6 days after noise; tizaterkib-treated mice had lower amounts. We also observed lower amounts of CD68 protein, a macrophage marker, in the cochlea of tizaterkib-treated mice compared to noise-alone animals. Previous studies have demonstrated that up to 95% of the immune cells in the cochlea are of monocyte/macrophage origin; therefore, we are proposing that tizaterkib mainly lowers the number of monocytes that are infiltrating into the cochlea and then differentiating into macrophages [36,37,45,79]. This was supported by the significant decrease in CD68 protein quantified in the total protein lysates of the cochleae of tizaterkib-treated mice compared to noise-alone mice on day 6 after noise exposure (Figure 8). We had another cohort of mice whose cochlea were analyzed 8 days following noise exposure and the number of CD45-positive cells were back to baseline levels in the noise-alone mouse cohort (Appendix A). This agrees with other studies that showed there was no longer an increased number of immune cells in the cochlea 7–8 days after noise exposure [36,37,82,83].

There are several interesting points raised when observing the immune cell data from this study. (1) ERK1/2 inhibition could be protecting mice from NIHL through lowering the number of infiltrating immune cells in the cochlea. This is a possible mechanism of protection that will further be explored to determine the role of immune cells in NIHL. (2) Our treatment protocol beginning 24 h after noise exposure and continuing for 3 days lines up with the same period of time in which most CD45-positive cells are infiltrating the cochlea. Immune cell infiltration starts to occur between 24 and 48 h after noise exposure and peaks around the 4-day mark, the same time point at which the mice are receiving treatments [37,45,79,82]. This could explain why 3 days of treatment confer a better protection from NIHL compared to 1 or 2 days of treatment (Figure 6 and Figure 7). (3) This study further implies that infiltrating immune cells could be a contributing factor to NIHL. There is a correlation between immune cell infiltration and hearing loss, as shown by the present study and others [35,37,42], but future studies will have to look deeper to determine whether an immune cell number above a specific threshold is a causative factor leading to hearing loss. 

Future studies will look at different immune cell populations to determine which ones are most affected by ERK inhibition following noise exposure. Even though it has been demonstrated that most immune cells in the cochlea are of monocyte/macrophage origin, neutrophils are another type of immune cell that are increased in the cochlea following noise insult [35,84,85]. Furthermore, we would like to determine exactly how ERK inhibition lowers the number of immune infiltrates in the cochlea. ERK has been shown to be involved in the cellular stress and death pathways, and inhibiting these pathways following noise could indirectly lower the number of immune cells [25,26,28]. Inhibiting cellular stress would lower the amount of pro-inflammatory molecules produced, such as cytokines, chemokines, and reactive oxygen species (ROS), which would then lower the number of immune cells infiltrating from the periphery [37,38,41,42]. The activation of ERK can also affect immune cell migration, so ERK inhibition could be directly inhibiting the migration of immune cells to the cochlea following noise insult [86,87]. It is interesting to note that ERK inhibitors may also play a key role in sensing damage levels and reducing the immune response in other post-mitotic cells outside the ear, such as in the kidney and neurodegenerative disorders of the joints and brain [88,89,90]. Finally, ERK1/2 inhibition has been shown to affect activity-regulated cytoskeleton-associated proteins (Arc) in neurons, which could be another possible mechanism of the protection from hearing loss seen with ERK inhibition [91,92]. There is some evidence that targeting cytoskeleton proteins can protect from hearing loss because Rac1 inhibition prevents cisplatin-induced OHC death in cochlear explants [17].

In summary, we show that tizaterkib, a highly specific ERK1/2 inhibitor, protects mice from NIHL at clinically relevant doses and in two mouse strains—FVB/NJ and C57BL/6 (KSR1 mice). The best time to start treatment is 24 h after noise exposure when the treatment continues for a total of 3 days. The drug protects both female and male mice with equal efficacy. This treatment regimen partially protects mice from cochlear synaptopathy, which is part of the tizaterkib treatment’s protective mechanism against NIHL. Additionally, tizaterkib protects mice from levels of noise exposure of 100 and 106 dB SPL, and the protective mechanism was suggested to be through the MAPK pathway and not other off-target pathways. Finally, ERK1/2 inhibition was shown to lower the number of CD45- and CD68-positive immune cells in the inner ear following noise exposure, which could be part of the protective mechanism of tizaterkib. This study further supports that targeting the MAPK pathway is a promising therapeutic strategy for mitigating NIHL, and the ERK1/2 inhibitor tizaterkib is an intriguing compound that needs to be further studied as a possible drug for alleviating NIHL in humans.

## 4. Materials and Methods

### 4.1. Ethics Statement

All animal procedures were approved by the Institutional Animal Care and Use Committee of Creighton University (IACUC) in accordance with policies established by the Animal Welfare Act (AWA) and Public Health Service (PHS).

### 4.2. Mouse Models

FVB/NJ mice were purchased from Jackson Laboratory and used as breeders in the Creighton University animal research facility. All FVB/NJ mice were 6–8 weeks old at the start of each experiment. KSR1 mice with a C57BL/6 background were a kind gift from Dr. Robert Lewis from the University of Nebraska Medical Center, Omaha, NE. KSR1 heterozygous mice were bred to obtain KSR1 KO and WT littermates. All KSR1 mice used for these experiments were 6–7 weeks old at the beginning of the experiment. All mice were cared for by the laboratory and animal facility staff.

### 4.3. Auditory Brainstem Response

ABR waveforms in anesthetized mice were recorded in a sound booth using subdermal needles positioned in the skull, below the pinna, and at the base of the tail, and the results were fed into a low-impedance Medusa digital biological amplifier system (RA4L; TDT; 20 dB gain). Mice were anesthetized using 500 mg/kg Avertin (2,2,2-Tribromoethanal, T4, 840-2; Sigma-Aldrich; St. Louis, MO, USA) with full anesthesia determined via a toe pinch. At the tested frequencies (8, 16, and 32 kHz), the stimulus intensity was reduced in 10 dB steps from 90 to 10 dB to determine their hearing threshold. ABR waveforms were averaged in response to 500 tone bursts, with the recorded signals filtered by a band-pass filter from 300 Hz to 3 kHz. The ABR threshold was determined by the presence of at least 3 of the 5 waveform peaks. Baseline ABR recordings were performed when mice were 7–8 weeks old and post-experimental recordings were performed 14 days after noise exposure. All beginning threshold values were between 10 and 40 dB at all tested frequencies. All thresholds were determined independently by two–three experimenters for each mouse, who were blind to the treatment the mice received. Threshold shifts were calculated by subtracting the pre-noise exposure recording from the post-noise exposure recording. ABR wave 1 amplitudes were measured as the difference between the peak of wave 1 and the noise floor of the ABR trace.

### 4.4. Distortion Product Otoacoustic Emission

Distortion product otoacoustic emissions were recorded in a sound booth while the mice were anesthetized. Mice were anesthetized using 500 mg/kg Avertin (2,2,2-Tribromoethanal, T4, 840-2; Sigma-Aldrich), with full anesthesia determined via a toe pinch. DPOAE measurements were recorded using a TDT RZ6 processor and BioSigTZ software (version 5.7.6). An ER10B+ microphone system was inserted into the ear canal in way that allowed for the path to the tympanic membrane to be unobstructed. DPOAE measurements occurred at 8, 16, and 32 kHz with an f2/f1 ratio of 1.2. Tone 1 was *0.909 of the center frequency and tone 2 was *1.09 of the center frequency. DPOAE data were recorded every 20.97 milliseconds and, on average, 512 times at each intensity level and frequency. At each tested frequency, the stimulus intensity was reduced in 10 dB steps starting at 90 dB and ending at 10 dB. The DPOAE threshold was determined by the presence of an emission above the noise floor. Baseline DPOAE recordings were performed when mice were 7–8 weeks old and post-experimental recordings occurred 14 days after noise exposure. Threshold shifts were calculated by subtracting the pre-noise exposure recording from the post-noise exposure recording.

### 4.5. Noise Exposure

Mice were placed in individual cages, in custom-made wire containers. System RZ6 (TDT; Alachua, FL, USA) equipment produced the sound stimulus, which was amplified using a 75 A power amplifier (Crown; Northridge, CA, USA). A JBL speaker delivered the sound to the mice in their individual chambers. The sound pressure level was calibrated using an NSRT-mk3 (Convergence instruments; Sherbrooke, Québec, Canada) microphone and all chambers were within 0.5 dB of each other to ensure equal noise exposure. Mice were exposed to 100- or 106-dB SPL noise for 2 h with an octave band noise of 8 to 16 kHz.

### 4.6. Tizaterkib Treatment

Tizaterkib (HY-111483) was purchased from MedChemExpress and administered to the mice via oral gavage. Tizaterkib was dissolved in 10% DMSO (D8418, Sigma-Aldrich), 5% Tween 80 (9005-65-6, MP Biomedicals; Santa Ana, CA, USA), 40% PEG 300 (192220010, ThermoFisher Scientific; Waltham, MA, USA), and 45% saline (0.9% NaCl). Mice were either administered 25, 5, 0.5, or 0.1 mg/kg/bw. Mice were treated both morning and night (12 h between treatments) for either 1, 2, or 3 whole days. The treatment began 45 min (Figure 1) before noise exposure or 24 (Figure 2, Figure 3, Figure 4, Figure 5, Figure 6, Figure 7 and Figure 8) or 48 h (Figure 6) after noise exposure. Mice were weighed periodically throughout the experimental protocol to monitor any drug toxicity and to ensure proper dosages were administered to the individual mice.

### 4.7. Ctbp2 Staining and Quantification

Mice were sacrificed following their post-experimental hearing tests and their cochleae were dissected and placed in 4% PFA. Their organs of Corti were micro-dissected and co-stained with anti-Ctbp2 (1:800; 612044, BD Transduction; Milpitas, CA, USA) and myosin VI (1:400; 25-6791, Proteus Biosciences; Redfern, New South Wales, Australia). Goat anti-rabbit Alexa Fluor 488 (1:400; A11034) and goat anti-mouse Alexa Fluor 647 (1:800; A32728) were purchased from Invitrogen as the secondary antibodies. Confocal imaging was performed using a Zeiss 700 upright scanning confocal microscope, with images taken with the 63× objective lens. Final images were achieved by taking a z stack image of the organ of Corti and processing it through the ZenBlack program (version 2.3). The number of Ctbp2 puncta were counted per IHC with a total of 12–14 IHCs analyzed per region counted. The total number of Ctbp2 puncta in each region were divided by the total number of IHCs in that region to determine the number of Ctbp2 puncta per IHC.

### 4.8. Cochlear Cryosectioning and CD45 Staining

FVB mice aged 6–8 weeks old were exposed to 100 dB SPL noise (8–16 kHz octave band) for 2 h. Mice were either treated with carrier alone or 0.5 mg/kg tizaterkib twice a day for 3 days beginning 24 h after noise exposure. Mice were then sacrificed 1 h after the last tizaterkib treatment, which was approximately 85 h (little over 3 ½ days) after noise exposure. Another set of mice were sacrificed 8 days after noise exposure to observe a different time point. Cochleae were extracted from the mice and placed in 4% PFA for 2–3 days. Cochleae were then decalcified in 120 nM EDTA for 2–3 days. Following decalcification, the cochleae were transferred to a 30% sucrose solution and kept at 4 °C overnight. The next day, samples were put into a solution of 30% sucrose and OCT compound (4583; Sakura; Torrance, CA, USA) for 4 h at 4 °C. The samples were then placed in OCT compound overnight at 4 °C. The next day, the cochlear tissues were oriented in within cryomolds containing OCT compound and frozen on dry ice. Frozen tissues were cut to a 10 μm thickness, captured on glass slides, and allowed to dry for several hours.

Cochlear cryosections were then blocked and permeabilized in a solution of 5% FBS and 0.2% triton X-100 in PBS. Tissues were stained overnight at 4 °C with mouse CD45 antibody (1:200; Af114, R&D Systems; Minneapolis, MN, USA). The next day, tissues were then stained for one and a half hours with Alexa Fluor 568 donkey anti-goat (1:400; A11057, Invitrogen; Waltham, MA, USA) and DAPI (1:1000; D1306, Invitrogen) to counterstain their nuclei. Tissues were mounted in Fluoromount g (00-4958-02, Invitrogen) and imaged using a Zeiss 700 upright confocal microscope. Post-acquisition images were analyzed using the IMARIS imaging software (version 10.0) and automatically quantified following intensity thresholding. CD45-positive cells were then cross-checked manually to ensure positive CD45 cells were co-stained with DAPI. Raw CD45 cell counts were normalized in the scala tympani region to the area of the region counted and averaged as CD45-positive cells per μm^2^.

### 4.9. Western Blotting

FVB mice aged 7–8 weeks old were exposed to 100 dB SPL noise (8–16 kHz) for 2 h and treated with carrier or 0.5 mg/kg tizaterkib twice a day for three days beginning 24 h after noise exposure. Mice were sacrificed 6 days after noise exposure and whole cochlear lysates were prepared in lysis buffer (9803; Cell Signaling; Danvers, MA, USA) after adding protease (cOmplete ULTRA Tablets 05892791001) and phosphatase (PhosSTOP 04906845001) inhibitors (Roche; Basel, Switzerland). The cochlea from each mouse were pooled together so each experimental group had tissues from 5 mice (10 cochlea). The lysates were centrifuged for 20 min at 16,000× *g* and 4 °C, and the supernatants were collected. Protein concentrations were determined with the BCA protein assay kit (23235, Thermo Fisher Scientific; Rockford, IL, USA). Twenty-five micrograms of total cell lysate were loaded on 10% SDS-polyacrylamide electrophoresis gel. After running the gel and transferring to a nitrocellulose membrane, the following antibodies were used for immunoblot analysis: anti-CD45 (1:1000; Af114, R&D Systems), anti-CD68 (1:1000; MCA1957, Bio-Rad; Hercules, CA, USA), and anti-GAPDH (1:10,000; AB181602; Cambridge, MA, USA). Rabbit anti-goat (1:4000; 31402, Invitrogen), goat anti-rat (1:4000; 31470, Invitrogen), and goat anti-rabbit (1:4000; P0448, Dako; Santa Clara, CA, USA) secondary antibodies were used. National Institute of Health (NIH) ImageJ software (version 1.54g) was used to quantify band intensities and these were recorded as a ratio with respect to the GAPDH loading control. Each pooled cochlear lysate was run 3 times.

### 4.10. Statistical Analysis

Statistics was performed using Prism (GraphPad Software; version 10.2.3). A two-way analysis of variance (ANOVA) or one-way ANOVA with a Bonferroni post hoc test was used to determine mean differences and statistical significance. For the ABR and DPOAE threshold shift graphs, the color of the asterisks indicates statistical significance of the treatment group of that same color compared to noise + carrier treated mice.

## Figures and Tables

**Figure 1 ijms-25-06305-f001:**
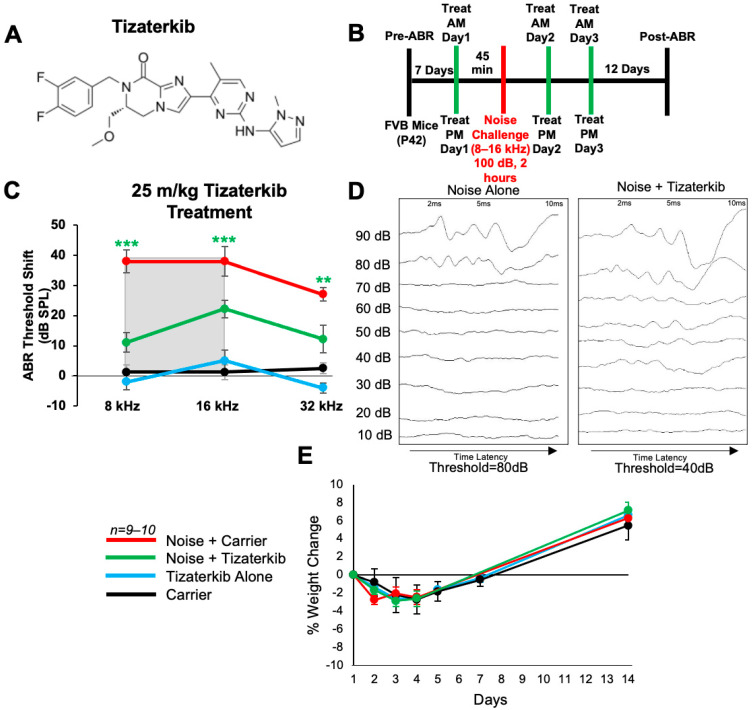
AZD0364 protects mice from noise-induced hearing loss when administered 45 min before noise exposure. (**A**) Molecular structure of tizaterkib. (**B**) Schedule of administration of noise exposure and tizaterkib treatments in FVB mice. Mice were given their first treatment of tizaterkib via an oral gavage 45 min before noise exposure. Mice were treated with the drug for a total of 3 days, twice a day, and exposed to noise once. (**C**) ABR threshold shifts following procedure in (**B**). Shaded region is the frequency range of the noise exposure. (**D**) Representative post-noise-exposure ABRs of noise-alone- and noise + tizaterkib-treated mice. (**E**) Percent weight change of different experimental cohorts throughout the 14-day protocol shown in (**B**). Noise + Carrier (red), noise + tizaterkib (green), tizaterkib alone (blue), and carrier (black). Data shown as means ± SEM; ** *p* < 0.01 and *** *p* < 0.001 compared to noise alone by two-way ANOVA with a Bonferroni post hoc test. The color of the asterisks indicates the statistical significance of the treatment group with that same color compared to noise + carrier treated mice. *n* = 9–10 mice.

**Figure 2 ijms-25-06305-f002:**
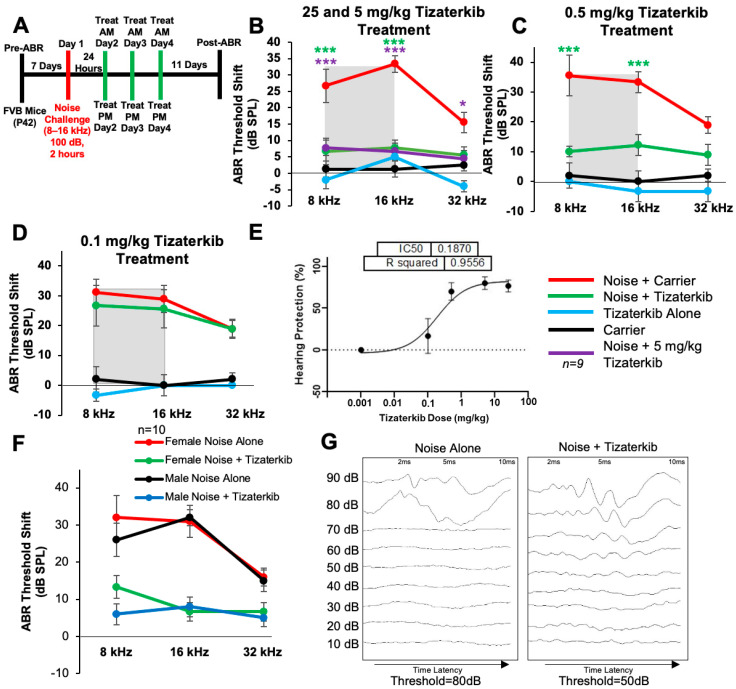
Tizaterkib protects mice from noise-induced hearing loss when administered 24 h after noise exposure. (**A**) Schedule of noise exposure and tizaterkib, which began 24 h after noise exposure. Mice were treated with varying concentrations of tizaterkib twice a day for 3 whole days. (**B**) ABR threshold shifts following procedure in (**A**), with 25 and 5 mg/kg tizaterkib given to separate groups. Shaded region is the frequency range of the noise exposure. (**C**) ABR threshold shifts following the procedure in (**A**) with 0.5 mg/kg administered to mice. (**D**) ABR threshold shifts following the procedure in (**A**) with the 0.1 mg/kg tizaterkib treatment. (**E**) Dose–response curve of tizaterkib protection from noise-induced hearing loss at 16 kHz, with 100% protection as a 0 dB SPL threshold shift. (**F**) ABR threshold shifts of males and females, graphed separately, that were treated with tizaterkib or carrier following the procedure in (**A**). (**G**) Representative ABR traces of noise-alone- and noise + tizaterkib-treated mice. Noise + carrier (red), noise + tizaterkib (green), noise + 5 mg/kg tizaterkib (purple), tizaterkib alone (blue), and carrier (black). Data shown as means ± SEM; * *p* < 0.05 and *** *p* < 0.001 compared to noise alone by two-way ANOVA with a Bonferroni post hoc test. The color of the asterisks indicates the statistical significance of the treatment group with that same color compared to noise + carrier treated mice. *n* = 9–10 mice.

**Figure 3 ijms-25-06305-f003:**
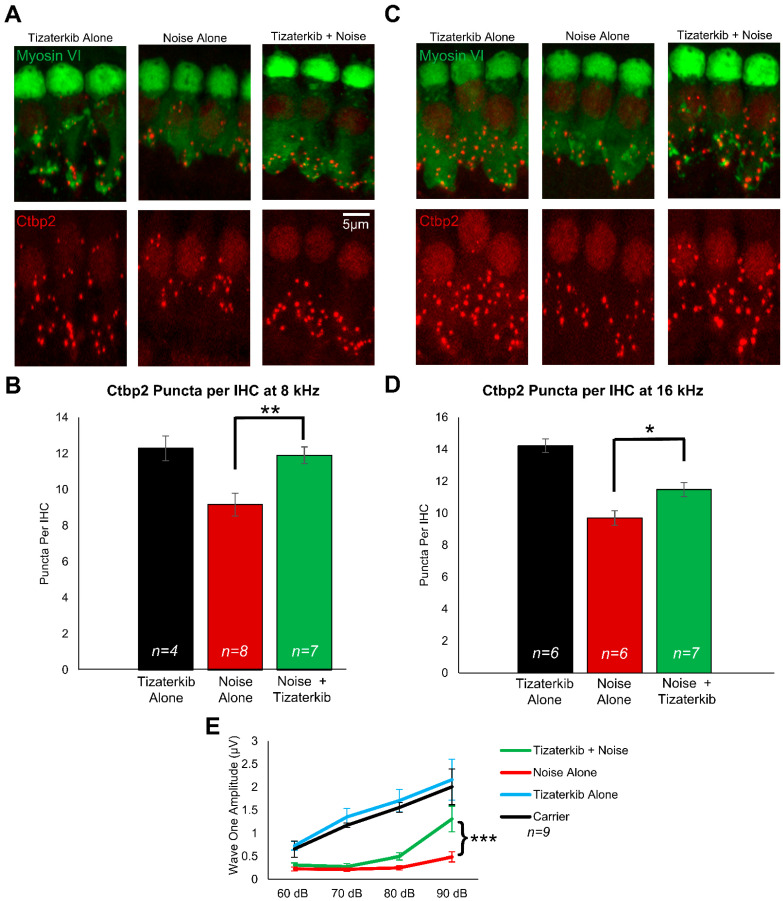
Tizaterkib protects mice from noise-induced synaptopathy in the 8 and 16 kHz regions. (**A**) Representative images of whole-mount cochlear sections stained with myosin VI (green) and Ctbp2 (red) in the 8 kHz region. (**B**) Number of Ctbp2 puncta per IHC in the 8 kHz region. (**C**) Representative images of whole-mount cochlear sections in the 16 kHz region. (**D**) Number of Ctbp2 puncta per IHC in the 16 kHz region. (**E**) ABR wave 1 amplitude for 16 kHz from the post experimental ABR recordings shown in Figure 2C. The wave 1 amplitude was measured from 60–90 dB. Data shown as means ± SEM; * *p* < 0.05, ** *p* < 0.01, and *** *p* < 0.001 compared to noise alone by one-way ANOVA with a Bonferroni post hoc test. Tizaterkib alone (blue), carrier (black), noise alone (red), noise + tizaterkib (green). *n* = 9 mice.

**Figure 4 ijms-25-06305-f004:**
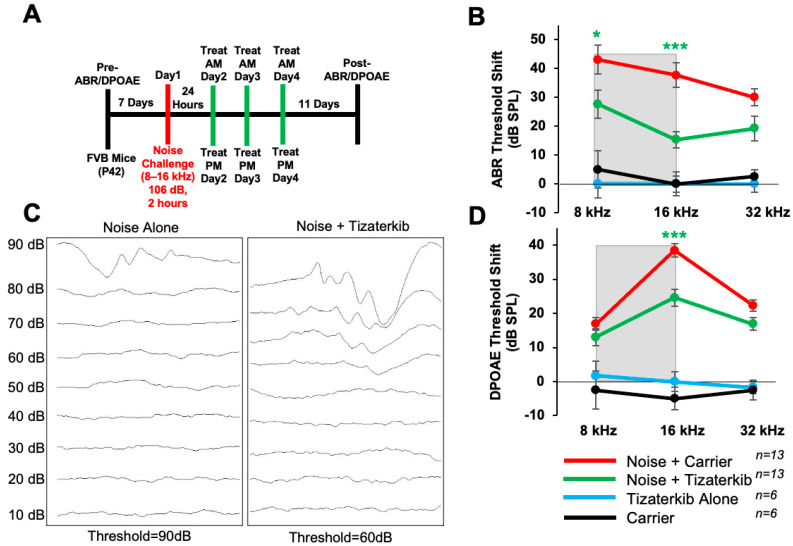
Tizaterkib protects from noise-induced hearing loss when mice are exposed to 106 dB. (**A**) Schedule of noise exposure and tizaterkib treatment. Mice were exposed to 106 dB SPL for 2 h and tizaterkib treatment started 24 h after noise exposure. Mice were treated for 3 whole days, twice a day. (**B**) ABR threshold shifts following the protocol in (**A**). Shaded region is the frequency range of the noise exposure. (**C**) Representative ABR traces of noise-alone- and noise + tizaterkib-treated mice following the 106 dB SPL noise exposure. (**D**) DPOAE threshold shifts following the protocol in (**A**). Noise + carrier (red), noise + tizaterkib (green), tizaterkib alone (blue), and carrier (black). Data shown as means ± SEM; * *p* < 0.05 and *** *p* < 0.001 compared to noise alone by two-way ANOVA with a Bonferroni post hoc test. The color of the asterisks indicates the statistical significance of the treatment group with that same color compared to noise + carrier treated mice. *n* = 13 mice.

**Figure 5 ijms-25-06305-f005:**
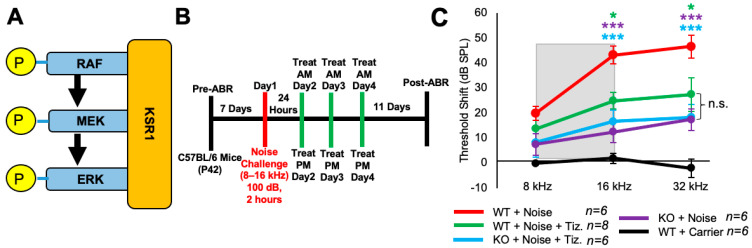
Tizaterkib treatment phenocopies the resistance to noise-induced hearing loss measured in the KSR1 KO mouse model. (**A**) KSR1 is a scaffolding protein for RAF, MEK, and ERK which enables the efficient transmission of MAPK signals. (**B**) Schedule of noise exposure and 5 mg/kg tizaterkib treatment in KSR1 WT and KO mice. Mice were exposed to 100 dB SPL for 2 h and tizaterkib treatment began 24 h after noise exposure. Mice were treated with tizaterkib or carrier twice a day for 3 whole days. (**C**) ABR threshold shifts following the protocol in (**B**). Shaded region is the frequency range of the noise exposure. WT + noise (red), KO + noise + tizaterkib (blue), WT + noise + tizaterkib (green), KO + noise (purple), and WT + carrier alone (black). Data shown as means ± SEM; * *p* < 0.05 and *** *p* < 0.001 compared to noise alone by two-way ANOVA with a Bonferroni post hoc test. The color of the asterisks indicates the statistical significance of the treatment group with that same color compared to noise + carrier treated mice. *n* = 5–6 mice.

**Figure 6 ijms-25-06305-f006:**
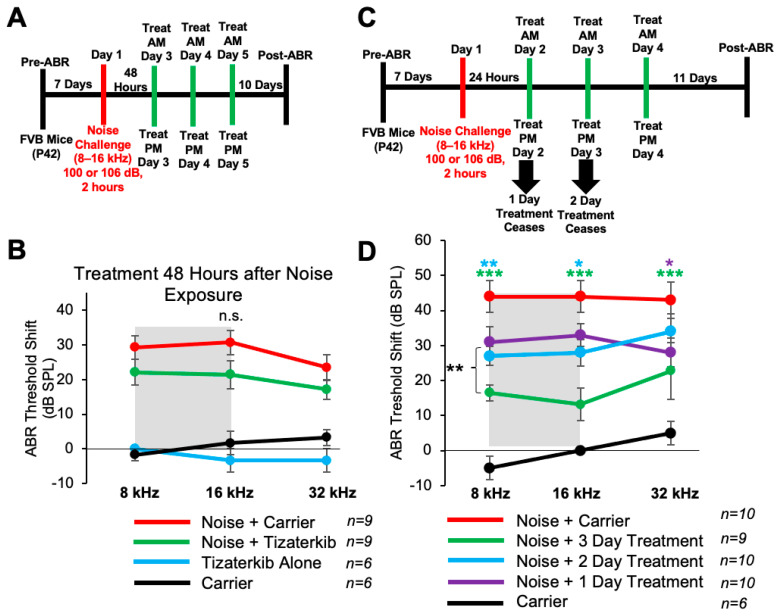
Three days of treatment beginning 24 h after noise exposure produces the optimal protection in terms of tizaterkib administration. (**A**) Schedule of administration for noise exposure and tizaterkib treatment. Treatment with tizaterkib began 48 h after noise exposure and mice were treated for 3 days, twice a day. (**B**) ABR threshold shifts following the protocol in (**A**). Shaded region is the frequency range of the noise exposure. Noise alone (red), noise + tizaterkib (green), carrier (black), and tizaterkib alone (blue). (**C**) Schedule of administration of noise exposure and tizaterkib treatment. Treatment began 24 h after noise exposure and one cohort was treated for 1 day, one cohort was treated for 2 days, and another cohort was treated for 3 days. (**D**) ABR threshold shifts following the protocol in (**C**). Noise alone (red), 1-day treatment + noise (purple), 2-day treatment + noise (blue), 3-day treatment + noise (green), carrier alone (black). Data shown as means ± SEM; * *p* < 0.05, ** *p* < 0.01, and *** *p* < 0.001 compared to noise alone by two-way ANOVA with a Bonferroni post hoc test. The color of the asterisks indicates statistical significance of the treatment group with respect to that same color compared to noise + carrier-treated mice. *n* = 6–10 mice.

**Figure 7 ijms-25-06305-f007:**
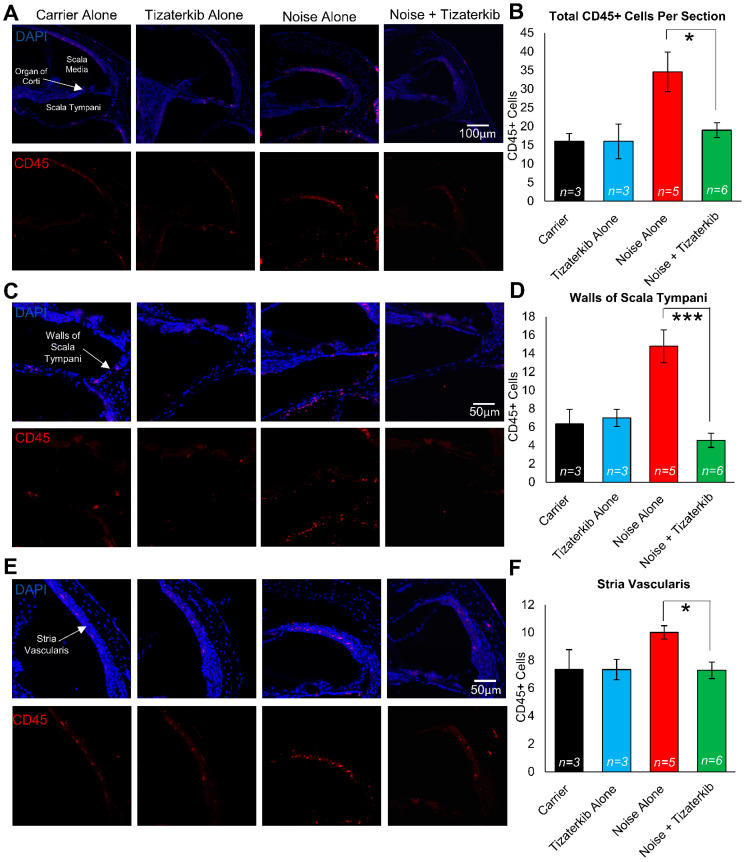
Tizaterkib treatment lowers the number of CD45-positive cells in the cochlea 4 days post noise exposure. (**A**) Representative low-magnification images of cochlear cryosections stained with CD45 (red) and DAPI (blue). The treatment protocol shown in Figure 2A was utilized and mice were sacrificed 4 days after noise exposure, 1 h after the final tizaterkib treatment. (**B**) Quantification of the CD45-positive cells in the cochlear sections in (**A**). (**C**) Higher magnification of the images shown in (**A**) of the scala tympani. (**D**) Quantification of CD45-positive cells in the walls of the scala tympani as presented in (**C**). (**E**) Representative images of cochlear cryosections of the stria vascularis following noise and tizaterkib treatment. (**F**) Quantification of CD45-positive cells per experimental group from the images in (**E**). Carrier (black), tizaterkib alone (blue), noise alone (red), noise + tizaterkib (green). Data shown as means ± SEM, * *p* < 0.05 and *** *p* < 0.001 compared to noise alone by one-way ANOVA with a Bonferroni post hoc test. *n* = 3–6 mice, with 3 sections each per mouse.

**Figure 8 ijms-25-06305-f008:**
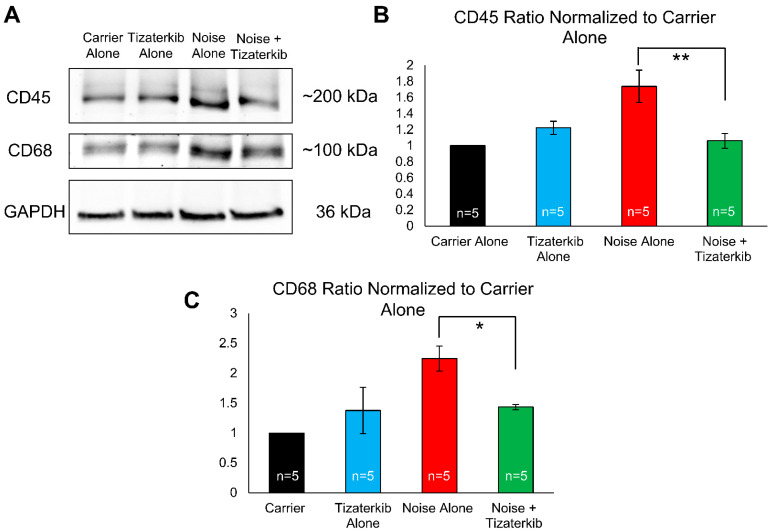
Tizaterkib treatment lowers the amount of CD45 and CD68 in the cochlea 6 days post noise exposure. (**A**) Western blots showing the amount of CD45 and CD68 in the cochlea following noise exposure and tizaterkib treatment. The same treatment protocol shown in Figure 2A was utilized and mice were sacrificed 6 days after noise exposure. (**B**) CD45/GAPDH ratio, normalized to the carrier-alone lane. Band intensities were measured using ImageJ software (version 1.54g). (**C**) CD68/GAPDH ratio, normalized to the carrier-alone lane. Data shown as means ± SEM, * *p* < 0.05 and ** *p* < 0.01 compared to noise alone by one-way ANOVA with a Bonferroni post hoc test. The experimental groups, from left to right, are as follows: carrier alone, tizaterkib alone, noise alone, and noise + tizaterkib. Each group had the cochleae from 5 mice (10 cochleae) pooled together to make the tissue lysate. *n* = 5.

## Data Availability

All data needed to evaluate the conclusions in this paper are present in the paper and Appendix A. Additional data related to this paper may be requested from the authors.

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
