# Peer review of "ERK1/2 Inhibition via the Oral Administration of Tizaterkib Alleviates Noise-Induced Hearing Loss While Tempering down the Immune Response"

_ijms, 2024, doi:10.3390/ijms25126305_

Round 1

Reviewer 1 Report

Comments and Suggestions for Authors

In this manuscript entitled “ERK1/2 Inhibition by Oral Administration of Tizaterkib Alleviates Noise-Induced Hearing Loss While Tempering Down the Immune Response,” the authors presented that Tizaterkib, whose clinical trial for cancer treatment is ongoing, is effective in alleviating noise-induced hearing loss in mice and decreasing the number of immune cells in mouse cochlea following noise exposure.

The results are clearly presented, and the manuscript is well-written. However, the reviewer would like to suggest some revisions to further improve the manuscript.

Major points:

[1] Figure 1C, 2B, 2C, 4B, 4D, 5C, 6D: The authors have included asterisks with certain colors. However, explanations for them are lacking. Do the asterisks mean that the groups shown by the same colored line in the graph are significantly different from the “Noise + Carrier” group shown by the red line?

[2] Although the authors mentioned that “There is no significant difference in threshold shifts between the WT mice treated with tizaterkib and either KO cohort exposed to noise (Fig. 5C),” it seems that there is a significant difference between the WT mice treated with tizaterkib and either KO cohort exposed to noise at 16 kHz.

[3] Fig.7B, D, and F and line 305: What does the ”n” of “n=3-6” mean? Does it refer to the number of animals or the number of sections? Please provide information about the number of sections and animals used in the experiments.

Minor points:

[1] Line 542: “45% 0.9% saline” is an odd description. Please provide the final concentration of saline.

Reviewer 2 Report

Comments and Suggestions for Authors

At the manuscript “ERK1/2 Inhibition by Oral Administration of Tizaterkib Alleviates Noise-Induced Hearing Loss While Tempering Down the Immune Response” by Drs. Richard D. Lutze et al. authors reported the new results of the study of the role of the kinases ERK1/2 for noise otoprotection by a newly developed, highly specific ERK1/2 inhibitor, Tizaterkib, in the animal models. Previously it was shown that Tizaterkib has high oral bioavailability and low predicted systemic toxicity in mice and humans. In the presented study authors performed dose-response measurements of Tizaterkib efficacy against permanent NIHL in adult FVB/NJ mice 17 and the minimum effective dose (0.5 mg/kg/bw), therapeutic index (>50), and window of opportunity (< 48 hours) were determined. Orally administrated drug   twice daily for 3 days, 24 hours 19 after 100 dB or 106 dB SPL 2hr noise exposure, at a dose equivalent to what is prescribed currently for humans in clinical trials, conferred average protection of 20-25 dB SPL in mice (both female and male).  Also it was shown that Tizaterkib can decrease the number of CD45 and CD68 23 positive immune cells in the mouse cochlea following noise exposure. Authors believe that   ERK1/2 kinases inhibition can be a promising strategy for treatment of NIHL.

 The authors did a great and impressive job. I have no substantive objections, but I just have some questions.

It has been shown that under certain conditions the Arc/Arg3.1 increases, accompanied simultaneously by increases in the extracellular signal-regulated kinase1/2 (pERK1/2):

doi: 10.1093/ijnp/pyv030

doi: 10.1523/JNEUROSCI.2410-07.2007

It seems that there is other indirect evidence of the connection between ERK-1/2 and Activity-regulated cytoskeleton-associated protein (Arc, Arc/Arg3.1).

 How realistic is the regulation of ERK-1/2 activity by feedback, depending on ARC expression?

Is it possible to use measurement of the expression level of the ARC peptide as an indirect indicator of the level of loss of neurons in the auditory system? (doi: 10.3389/fneur.2023.1201104)

Minor criticisms:

I would add the time/amplitude calibration bars to Figure 1 D and 2 G.

Excellent results obtained, presentation of a subject is systematic and comprehensive, I am happy to recommend the manuscript for the publication after minor corrections mentioned above.

Round 2

Reviewer 1 Report

Comments and Suggestions for Authors

The authors have responded to my concern satisfactory, and I think the revised paper now is suitable for publication.